# Unveiling Sex-Based Differences in the Effects of Alcohol Abuse: A Comprehensive Functional Meta-Analysis of Transcriptomic Studies

**DOI:** 10.3390/genes11091106

**Published:** 2020-09-21

**Authors:** Franc Casanova Ferrer, María Pascual, Marta R. Hidalgo, Pablo Malmierca-Merlo, Consuelo Guerri, Francisco García-García

**Affiliations:** 1Bioinformatics and Biostatistics Unit, Principe Felipe Research Center (CIPF), 46012 Valencia, Spain; fcasanova@cipf.es (F.C.F.); mhidalgo@cipf.es (M.R.H.); pmalmierca@cipf.es (P.M.-M.); 2Hospital Clinico Research Foundation, INCLIVA, 46010 Valencia, Spain; 3Department of Physiology, School of Medicine and Dentistry, University of Valencia, 46010 Valencia, Spain; mpascual@cipf.es; 4Department of Molecular and Cellular Pathology of Alcohol, Príncipe Felipe Research Center (CIPF), 46012 Valencia, Spain; cguerri@cipf.es; 5Atos Research Innovation (ARI), 28037 Madrid, Spain; 6Spanish National Bioinformatics Institute, ELIXIR-Spain (INB, ELIXIR-ES), 46012 Valencia, Spain

**Keywords:** alcohol use disorders, sex characteristics, meta-analysis, transcriptomics, functional profiling

## Abstract

The abuse of alcohol, one of the most popular psychoactive substances, can cause several pathological and psychological consequences, including alcohol use disorder (AUD). An impaired ability to stop or control alcohol intake despite adverse health or social consequences characterize AUD. While AUDs predominantly occur in men, growing evidence suggests the existence of distinct cognitive and biological consequences of alcohol dependence in women. The molecular and physiological mechanisms participating in these differential effects remain unknown. Transcriptomic technology permits the detection of the biological mechanisms responsible for such sex-based differences, which supports the subsequent development of novel personalized therapeutics to treat AUD. We conducted a systematic review and meta-analysis of transcriptomics studies regarding alcohol dependence in humans with representation from both sexes. For each study, we processed and analyzed transcriptomic data to obtain a functional profile of pathways and biological functions and then integrated the resulting data by meta-analysis to characterize any sex-based transcriptomic differences associated with AUD. Global results of the transcriptomic analysis revealed the association of decreased tissue regeneration, embryo malformations, altered intracellular transport, and increased rate of RNA and protein replacement with female AUD patients. Meanwhile, our analysis indicated that increased inflammatory response and blood pressure and a reduction in DNA repair capabilities are associated with male AUD patients. In summary, our functional meta-analysis of transcriptomic studies provides evidence for differential biological mechanisms of AUD patients of differing sex.

## 1. Introduction

Alcohol use disorder (AUD) is one of the most prevalent addictions in the world, according to data from the World Health Organization [1]. Alcohol abuse has been associated with more than 200 different health problems and pathologies [2], leading to high costs for social and health services across the globe. Alcohol consumption is also responsible for 6% of total deaths every year [1]. Amongst other effects, alcohol abuse can cause tolerance, physical dependence, and addiction.

AUD is a multifactorial disease that is influenced by both genetic and environmental factors [3]. Amongst those of environmental origin, social pressure, low socioeconomic level, and high stress during childhood are among the most relevant factors [4]. At the genetic level, there exist several genes that, individually, exhibit a modest influence in the probability of suffering from AUDs. For instance, genetic defects in enzymes involved in alcohol metabolism (Alcohol Dehydrogenase (ADH), Aldehyde Dehydrogenase (ALDH)) [5] and, in particular, a single mutation in the aldehyde dehydrogenase (ALDH2) gene, which codes for an enzyme that transforms acetaldehyde into to acetic acid, lead to acetaldehyde accumulation following alcohol consumption [6]. Thus, patients with specific variants of ADH suffer displeasing symptoms after low doses of alcohol, thereby discouraging alcohol consumption and, in turn, significantly reducing the chances of developing an addiction [7]. Epigenetic factors can also influence the development of addictions [8]. Sex can also influence the risk of developing AUDs [9] and can be seen as an environmental factor due to diverse cultural views regarding alcohol consumption in women and a biological/genetic factor [10]. Related studies on hormonal development have established that differences regarding AUD between men and women start appearing after puberty [9].

Due to the high prevalence of AUD, any added information regarding disease mechanisms and organismal effects may be of huge importance. A better understanding of alcohol addiction may also allow for the development of personalized therapies, and any studies aiming to understand any differences between men and women could implicate currently unknown genetic factors. A comprehensive understanding of specific factors would be relevant when developing new therapies or highlighting risk factors to be considered in patients of each sex.

As such, meta-analyses of transcriptomic studies conducted in the AUD field may provide some of the answers that we seek. The rapid development of the transcriptomics field has provided crucial information regarding how cell metabolism becomes altered in different situations, including the development of AUD [11]. Unfortunately, the obtained results generally lack a biological perspective, and the conclusions of these studies remain limited due to the methods usually employed in this field, such as differential expression analysis. However, the application of Gene Set Enrichment Analysis (GSEA) [12,13], which adds a layer of biological meaning to the results of the traditional analysis, and meta-analyses of similar studies, which provide statistically more powerful results, [14,15] can overcome these limitations. Altogether, the combination of these methods may provide answers regarding how sex influences AUD.

Given the utility of transcriptome analysis in understanding the molecular mechanisms of disease [16] and the impact of AUD on both social and health services, we aimed to identify those biological factors and mechanisms differentially affected in AUD patients of both sexes. We undertook a functional meta-analysis of transcriptomic data from studies found in public repositories that included samples of AUD patients of both sexes.

## 2. Materials and Methods

### 2.1. Systematic Review and Study Selection

The review and selection of studies were carried out between March to May 2020 in the Gene Expression Omnibus (GEO) [17] and ArrayExpress [18] public repositories. During this step, the guidelines of the PRISMA declaration for the elaboration of systematic revisions and meta-analysis were followed [15].

The search identified a range of transcriptomics studies related to human AUD. Keywords used during this step included but were not limited to: “transcriptomics”, “alcoholism”, “alcohol abuse”, “alcohol dependence”, “alcohol”, “ethanol”, “Alcohol Use Disorder” and “Homo sapiens.” From this set of studies, those that fulfilled the inclusion criteria were selected, which included data derived from RNA sequencing or genetic expression microarray platforms; the study had information about the sex of the subjects; the study had not been performed on cell lines; the study included a control group; and a minimum size of three subjects per experimental group. The normalized data of selected studies were downloaded using the R package GEOquery [19].

### 2.2. Bioinformatics Analysis Strategy

The same strategy was applied to the transcriptomic analysis of each selected study. This analysis included: data preprocessing, differential expression analysis, and functional enrichment analysis. Next, the functional results of all studies were integrated using meta-analysis techniques (Figure 1a depicts the bioinformatics analysis pipeline). Version 3.5.1 of R software [20] was used during the whole study. Every package and library used is detailed in Appendix A. Computer code is available at https://gitlab.com/ubb-cipf/metafunr.

### 2.3. Data Processing and Exploratory Analysis

Data preprocessing included the standardization of the nomenclature of the experimental group of each selected study, focusing on sex and diagnosis of AUD. The probe identifiers from the different platforms were also standardized. The Entrez code of the National Center for Biotechnology Information (NCBI) [21] was used for this step. Repeated probes were summarized using the median of their expression levels. An exploratory data analysis was then carried out (descriptive analysis of the expression levels, principal components analysis, and clustering analysis) to enable the identification of subjects with anomalous behavior or possible batch effects (Figure 1b,c).

### 2.4. Differential Expression Analysis and Functional Profiling

The analysis of differential expression levels between sexes was performed by using the R package limma [22]. For every gene, a linear model was adjusted. These models included the contrast to detect differences between women and men when comparing AUD and control groups:

(AUD Women - Control Women) - (AUD Men - Control Men)

*p*-values associated with the resulting statistics were adjusted using the Benjamini and Hochberg (BH) method [23]. Functional enrichment analysis was performed on the results of the differential expression analysis of each study. This functional profiling was performed using the GSEA method [12], implemented in the R package mdgsa [24]. This method detects functions that are overrepresented in groups of genes with a common expression profile. GSEA uses all the genes involved in the study, and genes are ordered by their level of differential expression and not only their significance; therefore, GSEA represents a more integrative approach to the study of gene expression. From this list of genes, ordered according to their differential expression, and the functional annotation of a given database (list of genes associated with each function), a logistic regression is adjusted for each function with the aim of explaining the relationship of a group of genes associated with a function and their expression level. In this type of regression, the relationship is quantified with the logarithm of the odds ratio (LOR), which compares the possibility that there is an association between a gene and a function against the possibility that there is no such relationship. Regarding the interpretation of the LOR, when a function has an LOR > 0, this indicates that the genes associated with that function have higher expression in women than men, while if a function has an LOR < 0, then the genes associated with that function will have higher expression in men than women. *p*-values obtained for every function were corrected again using the BH method. Functions with an adjusted *p*-value lower than 0.05 were considered statistically significant. The metabolic pathways of the Kyoto Encyclopedia of Genes and Genomes (KEGG) [25,26,27] and the Gene Ontology (GO) [28,29] were used for this functional enrichment analysis. GO terms were propagated separately for the three ontologies of this database: biological processes (BP), molecular functions (MF), and cellular components (CC).

For each ontology (BP, CC, and MF) and KEGG pathways, we analyzed the number of overrepresented elements shared by the studies. These results were graphically represented as UpSet plots [30] to depict the number of elements in common between the different sets.

### 2.5. Meta-Analysis

Results of the functional characterization of studies were integrated through a meta-analysis, which used the R packages metafor [31] and mdgsa [13]. First, the association with men and women of every KEGG pathway or GO term that appeared in at least two of the analyzed studies was determined. This process was performed using the odds ratio logarithms obtained using the DerSimonian and Laird (DL) method [32] available in the metafor package. This model enabled the detection of functions overrepresented in the set of analyzed studies, with better precision than that offered by the individual analysis previously performed, and thus offering greater statistical power. In the global estimation of the measured effect, the variability of the individual studies was incorporated, thereby granting greater statistical weight to studies whose values were less variable. The suitability of each analyzed studies was evaluated and confirmed with a heterogeneity study of the aforementioned indicators.

For each of the KEGG pathways and GO terms analyzed during the meta-analysis, the *p*-value, the LOR, and its confidence interval were calculated. *p*-values were adjusted using the BH method, and a particular term was considered significant if it had a *p*-value lower than 0.05. Significant terms with an LOR greater than 0 indicated an overrepresentation in women, while those with an LOR lower than 0 indicated an overrepresentation in men. Funnel plots and forest plots were used to evaluate the variability and the effect measure of every term in each one of the analyzed studies (Figure 1d,e). The significant results were represented graphically through dot plots and treemaps.

A total of 12,078 BP terms, 1723 CC terms, 4182 MF terms, and 229 KEGG pathways were evaluated during the meta-analysis.

### 2.6. Web Tools

The large volume of data and results generated in this work is freely available in the metafun-AUD web tool (https://bioinfo.cipf.es/metafun-AUD), which will allow users to review the results described in the manuscript and any other results of interest to researchers. The front-end was developed using the Bootstrap library. All graphics used in this tool were implemented with Plot.ly, except for the exploratory analysis cluster plot, which was generated with the ggplot2 package.

This easy-to-use resource is organized into five sections: (1) a quick summary of the results obtained with the analysis pipeline in each of the phases. Then, for each of the studies, the detailed results of (2) the exploratory analysis, (3) the differential expression, and (4) the functional characterization are shown. The user can interact with the tool through its graphics and search for specific information for a gene or function. Finally, in Section (5), indicators are shown for the significant functions identified in the meta-analysis that inform whether they are more active in women or men. Clicking on each indicator obtains the forest plot and funnel plot that explain the effect of each function in individual studies, as well as an evaluation of their variability.

## 3. Results

We have organized the results into three sections. The first describes which studies were assessed and selected in the systematic review, the second section demonstrates the results of the bioinformatic analysis of each of the selected studies (with (i) exploratory analysis, (ii) differential expression, and (iii) functional enrichment), while the third section summarizes the overall results of the studies on the differential functional profiling by sex.

### 3.1. Systematic Review and Study Selection

We identified 1416 studies that described illnesses/disorders related to alcohol abuse during the systematic review, of which we selected only 72 after refining the search. With this refined search, we avoided interference from studies related to other illnesses associated with alcohol use, such as fetal alcohol spectrum disorders, or unrelated studies that employed alcohol as a reagent or as part of growth media. Of these 72 studies, we only selected those transcriptomic studies of AUD performed in humans, thereby reducing the number of valid studies to 23. Most studies excluded at this point had been performed in animal models (e.g., rat and mouse), while we also discarded several studies with a methylome-based rather than transcriptomic approach, since we will subsequently need the same type of data in our bioinformatics strategy. Finally, we excluded those studies (74%) that lacked information regarding the sex of the subjects or had less than three subjects in any of their experimental groups. Remarkably, most of the excluded studies in this final step were due to the lack of a sex perspective. We also discarded studies without control subjects, which brought the number of applicable studies for meta-analysis down to four.

Altogether, the number of samples in these studies amounted to a total of 151 individuals: 49 control men, 50 AUD men, 27 control women, and 25 AUD women. In three of the four studies evaluated, the age of participants was available (mean and standard deviation in years): GSE44456, 58.44 (10.36); GSE49376, 56.5(9); GSE59206, 29.18 (7.74). Figure 2 depicts the flow diagram of the review system and study selection. The description of selected studies and the distribution of samples in each experimental group are detailed in Table 1 and Figure 3.

In both GSE44456 [33] and GSE49376 [34], samples were obtained from the New South Wales Tissue Resource Centre at the University of Sydney, Australia [35]. This institution has a track record of collaboration in studies regarding AUD, and the volunteers whose samples were used in these studies were diagnosed using the Diagnostic Instrument of Psychosis (DIP) screening instrument [36]. In GSE52553 [37], samples were obtained from the Collaborative Study on the Genetics of Alcoholism (COGA), which included up to 8000 individuals from the United States [38], and we only used the expression data from samples that had been grown in control conditions in this study. In GSE59206 [39], samples were obtained from the inhabitants of the community surrounding Yale University School of Medicine, USA. We only used the expression data from “baseline time point in neutral conditions” samples in our study. Of note, subjects in GSE59206 were recruited under the condition that they did not suffer from alcohol dependence according to the fourth version of the Diagnostic and statistical manual of mental disorders (DSM-IV); however, the original study notes that several subjects from the “Heavy Drinking” group engaged in binge drinking. Due to this caveat, the requirements to be part of said group (consumption of 15 or more standard alcoholic beverages per week in men and 8 or more standard alcoholic beverages per week in women), and the updated fifth version of DSM our experts used [40], we considered these subjects as part of the AUD group in our meta-analysis, even though they were not diagnosed with such an addiction in the original study.

### 3.2. Individual Analysis of the Studies

The initial exploratory analysis helped to pinpoint any bias caused by the batch effect in the studies corresponding to GSE44456 and GSE59206 (detailed results in the metafun-AUD web tool). We corrected the batch effect using the limma package [22], which incorporates a control variable in the linear model used in the differential expression, adjusting the part of variability that corresponds to this effect.

With the results of the differential expression analysis performed in each study, we also performed a functional enrichment analysis for every GO ontology and KEGG pathway. In this analysis, terms associated with one condition can be due to an overrepresentation of that term in the members of one sex when they suffer AUD, or the underrepresentation of the same term in the members of the opposite sex when they suffer AUD, and that could cause an indirect overrepresentation due to the relativistic nature of the results of the comparison performed.

Individual functional enrichment analysis of GO terms and KEGG pathways revealed the highly diverse nature of significant results among studies (Table 2). The relationship analysis of the significant functions using UpSet plots (all GO functions in Figure 4 and specific functions by ontology in Appendix A) indicated a low number of functions obtained by the simple intersection between studies. These parameters reinforce the implementation of functional meta-analysis, which avoids the loss of information and allows for the quantification of a combined measure of the activity for each function across all studies with greater precision than that provided by individual studies.

### 3.3. Meta-Analysis

We performed four groups of functional meta-analyses, one for each GO ontology and another one for the KEGG pathways using every term found in at least two of the selected studies. As a result of these meta-analyses, we indicated a total of 285 BP terms, 96 CC terms, 79 MF terms, and 6 KEGG pathways as significant (Table 3). These terms were overrepresented in either male of female AUD patients. The LOR values of these significant terms ranged between −0.62 and 0.98.

The functional groups with the highest overrepresentation for each sex have been summarized using treemaps and are shown in Appendix A.

Among the terms overrepresented in male AUD patients and thus underrepresented in female AUD patients, we found significance for several terms related to tissue growth and remodeling. Interestingly, these terms also specifically link to tissues derived from the mesoderm, and we also identified terms related to growth factors, adherens junctions, and collagen. We also uncovered significance for terms related to an increased innate immune and inflammatory response, including an increase in the secretion of Interleukin (IL)-1 and the migration of cells of the immune system. We also observed a general overrepresentation of terms related to the sense of smell, neural regeneration, triglyceride synthesis, response to angiotensin, activity on the plasma membrane, transcriptional regulation, and inhibition of proteolytic activity in male AUD patients (see Figure 5).

Among those terms overrepresented in female AUD patients and thus underrepresented in male AUD patients, we found significance in terms related to synaptic activity, vesicle formation, ciliary activity, melanosome organization, DNA reparation, protein transport, and growth factor inhibition. In this case, we also found significance with a considerable number of terms related with almost every step of gene expression, including nuclear and mitochondrial transcription, histone acetylation, RNA maturation involved in the formation of both tRNA and ribosomes, translation, and the degradation of both RNA and proteins (see Figure 5).

### 3.4. Metafun-AUD Web Tool

The Metafun-AUD web tool (https://bioinfo.cipf.es/metafun-AUD) contains information related to the four studies and 151 samples evaluated in this study. The portal includes fold-changes of genes and log odds ratios of functions and pathways for each evaluated study, which can be explored by users to identify profiles of interest.

We conducted a total of 18,212 meta-analyses. For each of the 466 significant functions, metafun-AUD demonstrated the global activation level by sex for all studies and the specific contribution of each study, using statistical indicators (log odds ratio, confidence interval, and *p*-value) and graphical representations by function (forest and funnel plots). This open resource aims to contribute to data sharing between researchers to aid the elaboration or interpretation of similar studies.

## 4. Discussion

Using a functional meta-analysis of transcriptomic data from different studies found in public repositories, we identified, for the first time, those biological mechanisms differentially affected in male and female AUD patients and obtained statistically significant results that more limited intersectional methods may have missed [15,32].

Even though sex differences in AUD have a proven biological basis, the most common scenario in studies includes few to no women. This situation has caused sex to be seldomly accounted for when studying the effects of AUD [41], and we observed this bias in most of those studies discarded during the study selection process. Including the perspective of sex in the present study of AUD provided a better characterization of the differences between male and female AUD patients besides its origin and should allow a better understanding of both risk and biological factors associated with AUD and the efficient adaptation of therapies and pharmacological treatments available [41].

We observed a remarkable lack of standardization among studies. To counter this unfortunately general problem, several experts have already suggested that studies follow four basic principles: Findable, Accessible, Interoperable, and Reusable, with studies that fulfill these requirements labeled as “FAIR” (Findable, Accessible, Interoperable, and Reusable) [42]. The application of these principles has already become common in projects of greater scope, which ensures the easier generation and divulgation of data [42,43].

The low number of studies selected represents a limitation with our systematic review; however, we preferred to keep strictly to our selection criteria to ensure comparability. We are also aware that the different types of tissue used in the studies and an unbalanced group of samples by sex in some cases decreases the statistical power. Given these points, and the knowledge that these studies are independent and derive from different environments, we treated this specific variability using a meta-analysis of random effects that detects those mechanisms with a similar pattern in the set of studies. Furthermore, we carried out a subsequent evaluation to confirm that no studies displayed a constant atypical pattern in functions and genes, ensuring the robustness of the obtained results.

The central nervous system is one of the organs most affected by chronic alcohol consumption. Studies in human postmortem brain samples have identified specific pattern of gene expression in brain of alcoholics when compared with controls [44,45]. More recently, transcriptome analysis from pre-frontal cortex of alcoholics further confirms the role of specific genes and biological pathways associated with alcoholism and AUD [46,47]. Nevertheless, these studies have been performed with male samples, and sex differences have not been described. In general, we found that female AUD patients displayed an increase in the activity of several neuronal and synaptic functions compared to male AUD patients. These results agree with studies, for example, by de la Monte et al. and Pfefferbaum et al., where an alteration in activity associated with an elevated generation of glial cells, which are significantly affected by alcohol and are significantly altered in our study [48,49]. However, female AUD patients exhibit worse transduction of olfactory signals and decreased neuronal regeneration in comparison to male AUD patients [50]. This finding suggests that cells of the brain in female AUD patients may attempt to counteract neuronal function and regeneration problems with an increase in activity, which could be related to the increased activity of the reward system observed in previous studies, thereby leading to a greater vulnerability to addiction. Indeed, studies have established sex-based differences in alcohol addiction in humans and experimental animals [51] and ethanol reward-seeking behaviors [52]. For instance, preclinical studies have consistently shown that under a variety of circumstances, female humans, rats, or mice drink significantly more ethanol than male counterparts [53]; however, genes associated with the reward system (e.g., c-Fos or -Fos B) do not exhibit different expression patterns between the sexes after ethanol exposure in rodents.

We also observed sex-based differences in the immune system response, which agrees with a previous study that described brain region-specific increases in microglial markers in human postmortem brain samples from moderate alcoholics [54]. For instance, we discovered increased activity and migration of several cell types related to the innate immune response in male AUD patients. This increase might relate to an augmented response upon the presence of pathogens and inflammation, and studies have related an increased inflammatory response to AUD [55,56], which might be related to a decrease in the production of tumor necrosis factor-α. However, the innate immune response may be depressed in female AUD patients, making them more vulnerable to infections. In agreement with the activation of the immune system in men, a recent study demonstrates that increased mean diffusivity in the brain gray matter of humans and rats undergoing chronic drinking associates with a robust decrease in extracellular space tortuosity induced by microglial activation, which could facilitate the dopamine pathways and contribute to the progressively enhanced addictive potency of alcohol [57]. It is interesting to note that gender differences have been observed in microglia cells in resting and developmental brain [58] and sexual differentiation of microglia and its impact on brain physiology and pathology has been shown [59]. Furthermore, in association with the difference in immune activity, we also observed an increase in hematopoietic function in male AUD patients in comparison to female AUD patients, which could be implicated in overall tissue growth and not only during embryonic development. This increase in tissue growth and inflammatory response may suggest that the effects of AUD make men especially vulnerable to some kinds of cancer in comparison to women, an idea supported by the decreased representation of terms related to DNA repair in male AUD patients [60].

We also observed that alcohol induces wide-ranging alterations related to tissue regeneration and scarring. Male AUD patients exhibited an increased platelet activity, which could relate to the observed increased response to several growth factors, endothelial proliferation, and epithelial differentiation. This finding suggests that female AUD patients suffer more severe inhibitory alterations in these processes, and thus their response to the presence of wounds is slower and less efficient. These functions may relate to the formation of adherens junctions or binding of cells to the extracellular matrix, which we observed to follow a pattern similar to the aforementioned functions. However, on a more specific level, male AUD patients suffer more significant melanosome degeneration and loss of pigmentary function than female AUD patients, which would suggest a less efficient performance of the function of the skin as a pigmentary barrier and thus the more rapid formation of skin lesions due to exposure to the sun.

At the cardiovascular level, we observed an augmented response to angiotensin in male AUD patients, suggesting a greater tendency to suffer high arterial pressure levels due to the vasoconstricting effects of angiotensin [61].

We also noted an overrepresentation of terms related to microtubules, including cilia structure and protein and vesicle transport, in female AUD patients. These results are similar to those recently published by Hitzemann et al., which observed an increased preference for alcohol consumption and alteration in both ciliary organization and extracellular matrix in female mice [62]. These functions may also relate to the increase in synaptic and melanosome organization already observed in women, but due to their importance in a variety of different functions, they could be involved in several alterations related to transport at a cellular level [63,64].

Of particular note, we observed the underrepresentation of several functions related with the development of tissues and organs in female AUD patients, which, together with terms related with embryonic development, agree with established knowledge regarding the negative impact of alcohol on pregnancy, especially in the development of nervous structures [65]. These alterations could also relate to changes to the extracellular matrix, which has a crucial function during embryonic development in cell migration and has been noted as an element more significantly affected by AUD in women compared to men. Indeed, chronic exposure to alcohol can cause a variety of problems in the female reproductive system, including abnormal menstrual cycles, a failure to ovulate, an increased risk of spontaneous abortions, and early menopause [66]. All the described malformations correspond with those observed in children diagnosed with fetal alcohol syndrome [67]. Additionally, in relation to cell metabolism in general, female AUD patients display an overrepresentation of terms related to the synthesis and degradation of several kinds of RNA. These alterations also include an increase in histone acetylation, RNA methylation, a general increase in nuclear and mitochondrial activity, and alterations in terms related to protein synthesis and degradation. Although the implications of these particular alterations are diverse and require in silico and in vitro confirmation, recent studies have linked alterations to transfer RNA metabolism to neurodevelopmental disorders [68].

Finally, in many of the observed processes and mechanisms, we observed the presence of terms implied in totally opposed functions; we hypothesize that this may represent an effort to compensate for the changes prompted by AUD.

In summary, our findings provide new insight into the complex biological processes and the differential profile of the biological mechanisms in male and female AUD patients. Said functions and pathways might be helpful to better understand sex-based differences in AUD, and their in-depth study could open the door to the development of more effective, personalized treatments for this pathology.

## 5. Conclusions

In conclusion, we provide further evidence for functional meta-analysis as a robust and efficient means of evaluating and integrating data derived from transcriptomic studies with differing approaches. Furthermore, the application of this method promotes the use of FAIR data in future biomedical studies. The strategy followed in this study fostered the detection and characterization of functional differences caused by sex in AUD-driven changes at a transcriptomic level. These alterations include decreased neuronal and tissue regeneration, malformations to the embryo during pregnancy, alterations related to intracellular transport, and the increased replacement rate of both RNA and proteins in female AUD patients. Meanwhile, male AUD patients displayed an increase in inflammatory responses and blood pressure and a decrease in the ability to repair DNA, which may relate to the increased risk of cancer.

These results confirm the utility of incorporating the perspective of sex into biomedical studies, thereby improving our understanding of AUD-related mechanisms in men and women, and generating relevant information for the development of efficient, personalized treatments.

## Figures and Tables

**Figure 1 genes-11-01106-f001:**
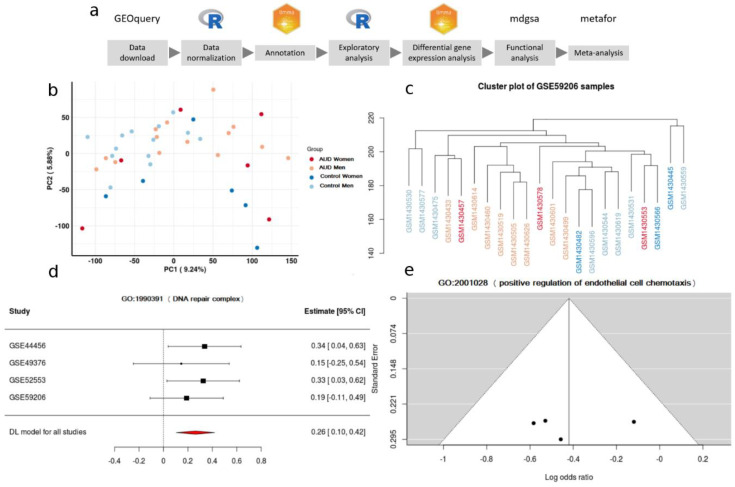
(**a**) Data-analysis workflow. (**b**) Principal Component Analysis plot in the GSE44456 study. (**c**) Clustering. (**d**) A forest plot of the GO:1900391, showing the LOR (log odds ratio) of each study and the global result. (**e**) Funnel plot of the GO:2001028; dots in the white area indicates the absence of bias and heterogeneity.

**Figure 2 genes-11-01106-f002:**
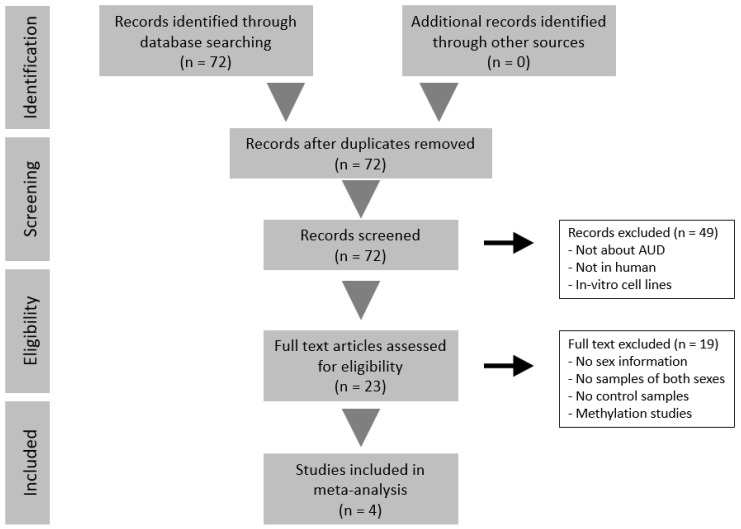
Flow diagram of the systematic review and selection of studies for meta-analysis according to PRISMA statement guidelines.

**Figure 3 genes-11-01106-f003:**
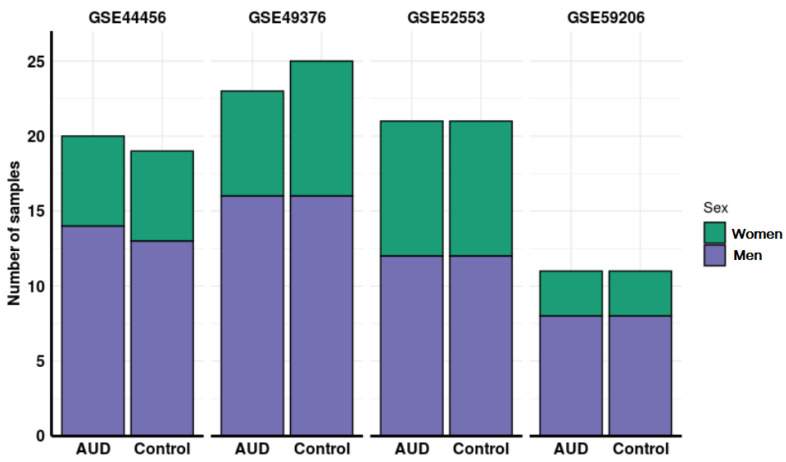
Distribution of samples by sex and addiction status in each of the studies analyzed.

**Figure 4 genes-11-01106-f004:**
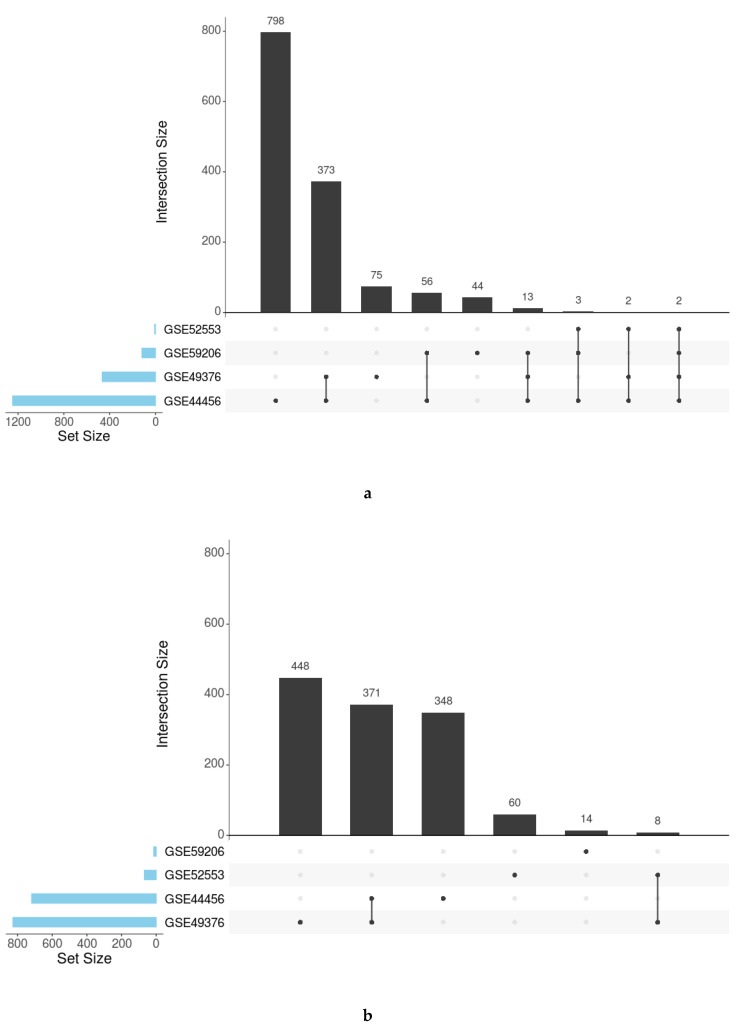
UpSet plots showing the number of common and specific GO functions in women (**a**) and men (**b**).

**Figure 5 genes-11-01106-f005:**
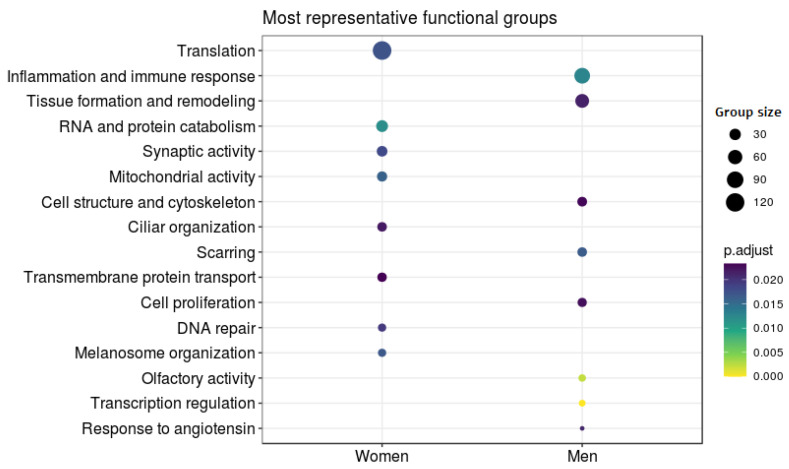
Differential Functional Profiling by Sex. The dot plot shows the functional groups with the greatest differential activity between the sexes. Each dot represents a biological function. Size indicates the number of genes involved in that function and color associated with the level of significance.

**Table 1 genes-11-01106-t001:** Studies selected for analysis after the systematic review. GEO (Gene Expression Omnibus) accession number, platform used, number of samples, sample tissue and citation number are included.

GEO Accession	Platform	Number of Samples	Sample Tissue	Citation
GSE44456 ^1^	GPL6244 Affymetrix Human Gene 1.0 ST Array	39	Hippocampus	McClintick, J. et al. [33]
GSE49376 ^2^	GPL10904 Illumina HumanHT-12 V4.0 expression beadchip	48	Dorsolateral prefrontal cortex	Xu, H. et al. [34]
GSE52553 ^3^	GPL570 Affymetrix Human Genome U133 Plus 2.0 Array	42	Immortalized lymphoblasts from blood samples	McClintick, J. et al. [37]
GSE59206 ^4^	GPL10558 Illumina HumanHT-12 V4.0 expression beadchip	22	Whole blood	Beech, R. et al. [39]

^1^https://www.ncbi.nlm.nih.gov/geo/query/acc.cgi?acc=GSE44456. ^2^https://www.ncbi.nlm.nih.gov/geo/query/acc.cgi?acc=GSE49376. ^3^https://www.ncbi.nlm.nih.gov/geo/query/acc.cgi?acc=GSE52553. ^4^https://www.ncbi.nlm.nih.gov/geo/query/acc.cgi?acc=GSE59206.

**Table 2 genes-11-01106-t002:** Number of significant GO terms and Kyoto Encyclopedia of Genes and Genomes (KEGG) pathways in each study after applying GSEA. Positive and negative LOR represent overrepresentation in women with AUD and men with AUD, respectively.

	GO Terms		KEGG Pathways	
Studies	Positive LOR	Negative LOR	Positive LOR	Negative LOR
GSE44456 ^1^	1208	703	39	25
GSE49376 ^2^	449	802	16	25
GSE52553 ^3^	7	66	0	2
GSE59206 ^4^	113	14	5	0

^1^https://www.ncbi.nlm.nih.gov/geo/query/acc.cgi?acc=GSE44456. ^2^https://www.ncbi.nlm.nih.gov/geo/query/acc.cgi?acc=GSE49376. ^3^https://www.ncbi.nlm.nih.gov/geo/query/acc.cgi?acc=GSE52553. ^4^https://www.ncbi.nlm.nih.gov/geo/query/acc.cgi?acc=GSE59206.

**Table 3 genes-11-01106-t003:** Number of significant GO terms and KEGG pathways resulting from each meta-analysis. Positive and negative LOR represent overrepresentation in female and male AUD patients, respectively.

Ontology/Database	Positive LOR	Negative LOR
Biological Processes	134	151
Cellular Components	73	23
Molecular Functions	55	24
KEGG pathways	5	1

## Data Availability

The data used for the analyses described in this work is publicly available at GEO (https://www.ncbi.nlm.nih.gov/geo/). The accession numbers of the GEO datasets downloaded are:
GSE44456 (https://www.ncbi.nlm.nih.gov/geo/query/acc.cgi?acc=GSE44456),GSE49376 (https://www.ncbi.nlm.nih.gov/geo/query/acc.cgi?acc=GSE49376),GSE52553 (https://www.ncbi.nlm.nih.gov/geo/query/acc.cgi?acc=GSE52553), GSE59206 (https://www.ncbi.nlm.nih.gov/geo/query/acc.cgi?acc=GSE59206). GSE44456 (https://www.ncbi.nlm.nih.gov/geo/query/acc.cgi?acc=GSE44456), GSE49376 (https://www.ncbi.nlm.nih.gov/geo/query/acc.cgi?acc=GSE49376), GSE52553 (https://www.ncbi.nlm.nih.gov/geo/query/acc.cgi?acc=GSE52553), GSE59206 (https://www.ncbi.nlm.nih.gov/geo/query/acc.cgi?acc=GSE59206).

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
