# Peer review of "Unveiling Sex-Based Differences in the Effects of Alcohol Abuse: A Comprehensive Functional Meta-Analysis of Transcriptomic Studies"

_genes, 2020, doi:10.3390/genes11091106_

Round 1
Reviewer 1 Report
The study by Ferrer et al. aggregates several studies assessing transcriptomics in men and women with AUD. The paper shows several AUD-driven changes at a transcriptomic level, with males showing -- as a summary -- more changes in pathways associated with neuroinflammation, blood pressure and DNA repair, whereas females showing alterations in tissue regeneration, pregnancy-related processes, intracellular transport, and RNA and protein replacement. Despite recent recommendations to enhance sex representation, women/female has traditionally been neglected in clinical and pre-clinical research, which in turns has severely delayed the discovery of treatments specifically tailored for them. The present study represents a step forward towards fixing this sad state of affairs. I thoroughly liked and enjoyed the present, which will be a nice addition to the AUD literature. I do have some suggestions and comments:
- The introduction is beautifully written. It introduces AUD as a whole, its distal genetic determinants and alterations in GABA and opioid transmitters as mediators as such determination. Sex differences are given appropriate treatment as well.
- A main feature of the study is it targeted studies published in public repositories, and among those chose those fulfilling certain criteria, such as having a control group. This is fine, yet it can reduce the scope and representativeness of the sample. For instance, it should be discussed which percentage of all published studies on the topic are found in these public repositories. How is the control of the data in those repositories? Moreover, a more complete description of the initial triage should be included. How many studies were initially gathered, and how many were discarded for not complying with each of the criteria. The material now in lines 195 to 201 and that in Fig. 2 provides some of this information but needs to be significantly expanded. Limitations associated with these issues should be given attention in the discussion. The fact that only 4 studies were finally evaluated merits these further comments.
- It is said that methylation studies were excluded. Usually meth studies have an initial transcriptome phase in which target genes are identified and then the meth analysis is conducted. Was not the case for the studies under analysis here (that then were excluded)?
- I congratulate the authors for the web tool developed at https://bioinfo.cipf.es/metafun-AUD
-Tree maps in Fig. 5 are difficult to read, please include them as supp. material and provide a link to bigger size pictures of them.
- Please provide a brief definition of LOD and the relevance of this odd-related indicator for readers not fully familiar with it. Please bear in mind the paper will be read, or should be read, by clinicians and pre-clinical researchers not fully familiar with "omics" jargon, that yet would fully benefit from knowing the differential sex effects described here to plan for their future research or use in their clinical work. If possible, include similar explanatory descriptions in other parts of the paper.
- The hypotheses put forward in lines 367 to 378 are quite intriguing, yet it is not clear to me which of the findings used to construct the hypothesis of "greater neural activity, greater activity of the reward system" are from the study itself and which from prior literature. Pre-clinical studies have consistently shown that female rats or mice drink significantly more ethanol than male counterparts, under a variety of circumstances. This information seems to support the hypothesis of the authors and could be added to the paragraph. On the contrary, to my knowledge at least, c-fos or delta-fos B preclinical studies that analyze induction of those genes in the reward system after ethanol exposure do not show a consistent sex-related pattern. Perhaps this info could be added as well to this paragraph.
- Lines 416-417: "which, together with terms related with embryonic development, suggest that alcohol may negatively impact pregnancy..." I understand the potential (i.e., "may") follows the style of prudently discussing the data, yet we already know that alcohol damages the fetus and impair several aspects of pregnancy (and some of the authors of this paper have made tremendous and wide known contributions to this issue). Please put the finding in a more confirmatory basis, in the style of the end of the paragraph.
Reviewer 2 Report
This manuscript describes a combined study-level and meta-analysis of sex differences in human AUD subjects. It also provides a web tool to further investigate the data from each or all of the studies. This is a very innovative and interactive tool and provides an easy way for investigators to confirm and display the manuscripts analysis and also ask some of their own innovative questions about specific genes. The authors are applauded for offering their R scripts for review. Overall, this is an innovative way to compile and present gene expression results from human tissue transcriptomic studies. However, there are a few weaknesses that could be addressed to further strengthen the scientific findings and aid in their interpretation.
Major concerns:
- Highly recommend having a link to the GEO datatsets both within the manuscript, and within the website/analysis tool.
- There is little to no description of the four studies used in these analyses. For each of the four GEO datasets, please provide how the AUD diagnosis was determined (if at all) and provide a description of the tissues used for the expression analyses. Perhaps, an additional column for tissue type in Table 1 would be helpful. If not all the studies used diagnosed AUD patients, please refrain from referring to the subjects as “AUD patients”.
- For study GSE59206, please indicate which data sets were used in your analysis. This study had three different conditions and several timepoints, but clearly, not all of them were used in your analyses.
- In the Methods, give a more complete description of the GSEA analysis. Briefly describe how it works and if a p-value cutoff for the DEGs input into the analyses was used.
- In the results, please describe the actions that were taken to account for batch effects.
- In the discussion, a few elements need further explanation. For example, there is no discussion of the fact that each study investigated here used different tissue types and that may significantly skew the resulting GO and KEGG results. Additionally, each study had different representation of female to male samples, bringing into question whether there is significant power to find sex differences. In one study, only 3 females per group were used, while 8 males/group were used. This was also in peripheral blood cells. Is it possible that inflammatory responses were found to be overrepresented in males simply because there were more male peripheral blood and lymphoblastoid samples used in the meta analysis? Please provide discussion of how the differing tissue types and differing representation of males:females for each of the studies may be affecting your overrepresentation meta-analyses
- In the Discussion, some results may be over interpreted. In Lines 415-425, the discussion of altered tissue development in offspring of AUD women is a bit of an over-statement. No reproductive tissue was assessed and the women were likely not pregnant at the time of tissue collection. The gene expression results are occurring within the women themselves. If you are proposing multi- or trans-generational transmission of these gene expression effects on future offspring, please support such statements with references. In Lines 407-409, the statement suggests that cardiovascular tissue was surveyed and no references were given to support the rest of the claim.
- The GSE datasets listed in the supplementary figures are not the same as those in the manuscript.
Minor concerns:
Fig 1b, the data set used for this PCA is not stated.
